# Flexible Fiber Fabric for FRP–Concrete Connection of Thin Hybrid Slabs

**DOI:** 10.3390/polym13172862

**Published:** 2021-08-26

**Authors:** Amir Mahboob, Lluís Gil, Ernest Bernat-Maso, Amir Reza Eskenati

**Affiliations:** 1Strength of Materials and Structural Engineering Department, Polytechnic University of Catalonia, C/Colom 11, TR45, 08222 Terrassa, Spain; lluis.gil@upc.edu (L.G.); ernest.bernat@upc.edu (E.B.-M.); amir.reza.eskenati@upc.edu (A.R.E.); 2Serra Húnter Fellow, 08222 Terrassa, Spain

**Keywords:** CFRP, concrete, hybrid beams, experimental, bending test, modal analysis, shear connection

## Abstract

In order to combat corrosion issues, several studies on progressively replacing steel reinforcement elements with composite ones have been conducted in recent years. Hybrid steel–concrete thin slabs in which the steel acts as formwork are also candidates for update in the coming years. Achieving a reliable connection between fiber-reinforced polymer (FRP) and cast-in-place concrete is key to promoting this technology. This study analyzed different connection systems and proposes the novel approach of embedding a flexible fiber fabric as a superficially distributed connector between concrete and FRP. Eight specimens with four different connection strategies were tested using an experimental modal analysis and a quasi-static three-point bending test. The impact of the connection system on the vibrational response, flexural ultimate load, moment response, neutral axis position, shear and dissipated energy was obtained and compared. The results show that the use of an embedded mesh increases the frictional mechanism and produces the best performance in terms of load-bearing capacity and ductility.

## 1. Introduction

Traditional reinforced concrete (RC) structures exhibit aging problems related to the corrosion of the steel reinforcement, and there have been many recent studies into strengthening and retrofitting these structures using composite materials. 

Recently, the FRP–concrete concept was used in manufacturing hybrid slab [1], column repairing [2] and a wall system [3]. Some of the advantages of these hybrid elements are good flexural stiffness, a high load-bearing capacity and excellent corrosion resistance. 

There are two main typologies of FRP–concrete hybrid structures. In the first type, FRP is used as a tubular or box envelope for concrete, whereas in the second and more studied type, FRP is presented as a girder or a pultruded profile connected to a top concrete section to aid bending resistance. Regarding the first approach, several researchers have studied the bending resistance design of hybrid structures based on the box system. Yang et al. [4] performed tests over longitudinal pultruded GFRP box-profile and transverse steel stirrups. Some GFRP shear connectors (rib and bolted) enhanced the bonding between the GFRP profile and the concrete portion. In addition, Zhang et al. [5] experimentally studied a box-type profile using wet bonding (WB) and FRP shear key (SK) connections. Moreover, Liu et al. [6] tested pultruded GFRP three-cell box-section and concrete slabs for pedestrian bridges, while Dagher et al. [7] worked on solutions with tubular FRP filled with concrete for the same application. 

Far more research can be found regarding the second hybrid FRP–concrete typology, which is based on connecting FRP pultruded profiles to upper concrete sections to generate hybrid beams. One of the pioneering works on hybrid composite single beams is by Hulatt et al. [8], in which manufactured hollow beams of GFRP webs, concrete heads and carbon FRP (CFRP) plies in the bottom flange were studied. Along the same lines, T-shaped beams combining GFRP and steel reinforcement for girders were tested by Wu et al. [9]. Zhang et al. [10] tested I-shaped pultruded GFRP profiles with the web strengthened by externally bonded CFRP sheets, comparing both common concrete and ultra-high concrete (UHPC) cases. Nordin et al. [11] considered GFRP I-beams reinforced with CFRP in the bottom flange connected to a concrete head, while Zou et al. [12] investigated the shear behavior of FRP–concrete composite sections using bending tests.

Regarding more specific tests on the connection between FRP profiles and concrete parts, Gonilha et al. (2014) [13] performed several push-out tests and simulations of hybrid systems for a girder with I-shaped GFRP beams and concrete. Six specimens connected with steel bolts and/or bonding were considered in Koaik et al. [14], and I-shaped pultruded GFRP bolt-connected profiles with partial interaction can be found in Neagoe et al. [15]. The specific performance of FRP–concrete sections was also experimentally analyzed by Zou et al. [16]. These authors developed an experimental study of bolted hybrid FRP–ultra-high-performance concrete (UHPC). They performed 14 push-out tests with varying parameters, such as bolt strength, bolt diameter and bolt-embedding length. Similarly, Etim et al. [17] developed several push-out tests to investigate the shear transmission between normal concrete and GFRP profiles. However, analytical methods have not yet been fully developed enough to precisely predict the shear capacity of the composite sections due to the complexity of the FRP–concrete connection. A specific attempt at GFRP and concrete hybrid beams can be found in Neagoe and Gil [18]. To complement the modeling branch, there are some numerical approaches, such as Gong et al. [19], who evaluated the composite action of a hybrid beam of FRP and fiber-reinforced polymer–concrete. These authors assumed that only shear connectors provide composite action between the FRP and concrete components. A more applied numerical study, as the one presented in Muc et al., was aimed at the design of a GFRP bridge girder combined with concrete [20]. Real applications of this type of FRP beam and concrete hybrid structures can be found in Jurkiewiez et al. [21], who tested a GFRP and concrete footbridge until failure. In this case, the connection between the slab and the GFRP profiles was a mixed bonded/bolted connection. Along the same lines, a real-size bridge with a hybrid solution of GFRP and CFRP with a concrete slab and a span of 21 m was tested in Siwowski et al. [22], and the details of a girder are in [23]. 

As can be observed from the literature review, most of the developments in hybrid FRP–concrete structures have been oriented toward implementing beam-like solutions. There are few contributions to research on plane slab typology, in which FRP can even be used as formwork. On this topic, the work by Gai et al. [24] has to be highlighted as it investigated the use of GFRP formwork for concrete floor slabs. The concept of using FRP as formwork was presented by one of the pioneering studies in Hall and Mottram [25] in which a combination of pultruded GFRP profiles as permanent formwork for concrete was studied. Therefore, there is a comparative lack of research in the area of hybrid FRP–concrete slabs that needs to be covered.

In addition, it can be seen that the connection systems between FRP and concrete always rely on epoxy-bonded solutions for previously cast concrete, or on mechanical punctual connectors for cast-in-place concrete. Accordingly, a novel connection system based on embedding a flexible fiber fabric bonded to FRP and mechanically interlocking it with concrete is proposed, tested and analyzed. This is an entirely new technology, to the best of our knowledge, with the closest research considering fiber fabrics together with FRP profiles being found in the work by Sutter et al. [26], who tested hollow TRC beams reinforced with CFRP and embedded in concrete and mathematically analyzed them [27].

The current work focuses on the description and mechanical characterization of the structural response of a novel hybrid slab system made with a thin CFRP sheet, which is also the formwork, and a concrete block that is connected through a flexible fiber fabric. An additional connection using particle-based frictional enhancement is also considered and compared. Among the mechanical characterization tasks, the experimental modal analysis is aimed at qualitatively relating the vibrational response with the particular studied typologies of FRP–concrete connection systems. In addition, three-point bending tests are presented to analyze the load-bearing capacity, strain distribution, evolution of the position of the neutral axis and the dissipated energy, among other parameters. 

The study presented herein analyses, for the first time, the effect of a flexible fabric mesh as a connector between CFRP sheet and concrete, for producing hybrid structures aimed to support bending efforts. The potential benefits of this study include (i) opening the possibility of replacing thin steel sheets with FRP sheets, avoiding the corrosion problems typical of aggressive environments; (ii) characterizing a connection technology based on distributing the load-transferring mechanism to be more compatible with FRP materials whose mechanical connection is always controversial; and (iii) providing the order of magnitude of the load-bearing capacity of thin slabs produced with this novel connection based on flexible fiber fabric. 

## 2. Experimental Program

### 2.1. Materials 

The materials and the manufacturing process can be seen in Figure 1 from a to d and from e to g, respectively. 

#### 2.1.1. Concrete 

All specimens were cast using commercial dry concrete composed of 300 kg/m^3^ of Portland cement with continuous 0 to 12 mm of limestone aggregates and plasticizer, and mixed with a water/cement ratio of 0.6 reaching a compressive strength below 25 MPa. This intended low compressive strength was designed to represent a poor cast-in-place material with the future aim of being combined with a natural composite with a lower performance than the CFRP sheets (see Section 2.1.2) used in this research. In addition, this lower concrete quality helped us to develop and study the failure mechanisms involving concrete damage with regard to the studied novel connection system. 

The strength of the concrete was estimated using a non-destructive Schmidt impact hammer. Six repetitions were conducted on each specimen to estimate an average value of the compressive strength according to norms [28,29]. The results for each specimen are summarized in Table 1. 

#### 2.1.2. CFRP Sheet 

CFRP sheets were manufactured using a hand layout process on a mold of thin steel sheet typically used for roofing (Figure 1f). The fabric was unidirectional carbon fiber Masterbrace FIB 300/50 (Figure 1b). The laminate consisted of three plies with an orientation of 0°/90°/0°. Epoxy resin (Resoltech 1200 + 1204) (Figure 1e) was brushed over every ply using a fibers/resin rate by weight of 1. Resoltech 1200 + 1204 is a wet layup epoxy resin with no reactive diluents that cures at ambient temperature (gel time over 8 h). After impregnating the fibers, a counter mold was placed over it and a distributed weight of 25 kg was applied to it to guarantee uniform impregnation. The CFRP were left to cure for 3 days before unmolding. Tensile tests were conducted on eight longitudinal coupons extracted from the manufactured CFRP sheets according to ASTM [30]. The modulus of elasticity in the longitudinal direction was 45.55 GPa and the ultimate tensile strength in the same direction was 1120 MPa.

#### 2.1.3. Mesh

A flexible mesh made of glass fibers (MapeGrid (G220)) was used to connect the CFRP sheet with the concrete in the proposed novel approach. This mesh had a weight of 225 g/m^2^ and an average tensile strength of 45 kN/m (Figure 1a). The fibers were distributed in a grid of 25 × 25 mm to allow the mechanical interlocking of the cast-in-place concrete. Moreover, the grid was bonded to the CFRP sheet on the top flat surfaces using carbon fiber strips Figure 1g) and the same resin (Resoltech 1200 + 1204). Mesh was placed aligning the grid with the nerves of the slab in some specimens, while in others, the relative orientation between the grid and the nerves was 45°. 

#### 2.1.4. Sand and Gravel

Solid particles of different sizes were used to increase the friction interface between the CFRP sheet and the concrete (Figure 1c,d). The sand particles had a diameter of 0–4 mm, and the gravel particles had a diameter of 5–12 mm. The particles were bonded (Figure 1g) by brushing the inner bottom surface of the CFRP with Resoltech 1204, dropping them freely, curing them in environmental conditions for 3 days and removing the unbonded particles with a vacuum cleaner. 

### 2.2. Specimens

Eight hybrid slabs were manufactured with four different connection systems used to study the flexural performance of the slabs. In particular, the research was oriented to comparatively investigate the performance of: (i) mechanical connection provided by aggregates with different sizes bonded to the inner surface of CFRP modifying the roughness of the CFRP–concrete interface; (ii) the mechanical connection provided by a flexible fiber FRP mesh that was bonded to the CFRP sheet and that interlocked concrete into their gaps; and (iii) the orientation of the connection fiber mesh and its relationship with load transfer mechanisms. To provide comprehensive data according with these study aims, specimens were defined so to be able to compare the impact of including (G-S-SM) or not including (G-SM) small-size aggregate particles. It was also possible to study the impact of using (G-SM) or not using (G) the proposed novel connection based on flexible fiber fabric. Finally, it was also possible to analyze the impact of the fabric orientation by comparing specimens with straight mesh (G-S-SM) and inclined mesh (G-S-IM). Other parameters like the number of fabric layers, the material of the fabric, the shape of the thin FRP sheet or the type of adhesive used to bond aggregates and fabric with CFRP were not considered in this first study aimed to provide a global assessment of the novel use of a flexible fabric to connect FRP and concrete. All these parameters, among others, should be analyzed in future works. 

The labeling of the samples describes the connection system and the numbering of the specimens. Numbers at the end of the labeling (1 or 2) define the specimen repetition. Letters define the connection system: G- for bonded gravel particles; G-SM for the combination of bonded gravel particles and straight (0°) oriented glass fiber mesh; G-S-SM for the same as G-SM with the addition of sand particles to the gravel ones; and G-S-IM for the analogous case but with the mesh oriented at 45° with respect to the CFRP sheet nerves. The variety of specimens is defined in Table 1. All were 2000 mm in length, 400 mm in width and 75 mm in height. Figure 2 shows the different connection systems.

## 3. Methodology

In order to study the different connection systems and to assess the novel proposal of using a flexible fiber fabric as a FRP–concrete connector, two types of tests were defined. First of all, experimental modal analysis was aimed to assess the performance of the studied connections in a non-destructive method with the final goal of providing a qualitative comparison. In addition, destructive bending tests were also conducted to provide quantitative data to compare the different studied connection systems. Direct results of these bending tests such as force–displacement curves, CFRP–concrete relative sliding and strain distribution are aimed to describe the mechanical response and the failure mechanism from a global point of view. In addition, some calculated variables are used to provide clear evidences of the failure mechanism (position of the neutral axis), practical information of the equivalent shear transferring capability (FRP–concrete equivalent shear stress) or to assess the ductility of the studied connections (external energy). 

Experimental methods are described in detail in this section.

### 3.1. Experimental Modal Analysis 

From the hypothesis that the connection system affects the shear transmission between the concrete block and the CFRP sheet, it was expected that the vibrational information would also depend on this connection. Experimental modal analysis (EMA) tests were performed to estimate the vibrational modes, frequencies and damping ratios of the specimens with the different proposed connection systems. The main aim was to qualitatively detect the differences between the connection systems on the vibrational responses. This information could be used to pave the way for future research on the non-destructive assessment of the connection systems of thin hybrid CFRP–concrete slabs, although it is outside the scope of the current research.

Specimens were placed as a simply supported beams with a free span of 1800 mm. Tests were conducted using impulse excitation based on ISO standard 7626-5 [31]. An impact hammer was used to excite the structure at different points (52 points, see Figure 3) controlling the applied force for long periods, whereas a uniaxial accelerometer was used to register the acceleration caused by impacts on a fixed point (Figure 3). The procedure followed a multiple-input, single-output (MISO) approach. Transient time weighting was used to acquire impact data, and exponential time weighting was used to acquire acceleration data, excluding the response forced by the hammer during the impact time, and using only the free vibration data. This strategy was successfully used for the analysis of CFRP reinforcement on masonry walls in previous research [32], where the full post-processing procedure was further detailed. 

### 3.2. Bending Test 

Three-point bending tests were performed on simply supported specimens to assess their performance with regard to bending and shear concomitance. Figure 4 shows the test set-p from the longitudinal (a) and transverse view (b). The free span was 1800 mm and the slab was simply supported on 50 mm diameter steel rollers. The load was indirectly applied using displacement control at a rate of 1 mm/min until failure. An oleo-hydraulic actuator of 50 kN force range and 150 mm displacement range equipped with a load cell and an LVDT was used for this purpose. A steel HEB120 profile was used as a loading tool to distribute the load along the slab width. The vertical displacement was also externally measured in the middle span cross-section with two 100 mm range 0.2% linearity potentiometers set at each side. In addition, two external LVDTs with a 20 mm range were used to precisely measure the relative displacement between the CFRP sheet and the concrete block at each slab end. These were supported on the CFRP edge and measured the relative movement of the concrete block with respect to this edge of the CFRP sheet. Finally, two strain gauges of 350 ohms resistance connected with a 4-wire configuration and temperature-compensated for composites were installed on the external face of the CFRP sheet at the mid-span position and at two different heights, as observed in Figure 4b. 

## 4. Test Results 

### 4.1. Modal Analysis

The modal shapes and associated frequencies (ω) and damping ratios (ζ) were experimentally obtained via modal analysis for each specimen. Table 2 summarizes the results of the vibrational response for all specimens. The third bending mode was the first one detected for all specimens; thus, it was used for comparison purposes. 

### 4.2. Bending Test Results

#### 4.2.1. Load Transfer Mechanism and Failure Mode

Different mechanisms for load transfer are supposed to be developed depending on the analyzed specimen. First of all, aggregates (gravel and sand) bonded to the inner surface of the bottom plates of the CFRP sheets are providing a rigid mechanical interlocking with the concrete. This connection system allowed a full load transfer between concrete and CFRP sheet at the beginning of the test up to the failure of this particular connection. The fragile failure of this load transfer mechanism started at the central section (greatest bending moment) and extended to the extremes. Hence, there was little contribution of the aggregates to any partial connection mechanism. Additionally, it has to be noticed that sand particles might not increase the mechanical interlocking as efficiently as particles of larger size such as gravels. According to the empirical evidences of the tests, the little size of sand particles (less than 4 mm) might determine a weaker sliding plane in the CFRP sheet–concrete interface compared to gravel aggregates. 

Regarding the load transfer mechanism of the fiber mesh, two cases have to be distinguished. Firstly, for straight mesh (G-SM and G-S-SM specimens), the grid geometry provided a flexible interlocking mechanism based on the shear deformation of the transversal fiber tows. The mesh was bonded to the CFRP sheet every 170 mm (top hat) and mechanically connected with concrete on every grid gap. The failure mechanism was based, in this case, in the shear progressive breaking of the transversal fiber tows, which started at the central section (between top hats) and progressed faster to the extreme edges (bonded). Full FRP mesh–concrete interaction and load transfer capability was observed up to this first breaking point. After shear breaking started, vertical deformation of bending specimens increased but load experienced a fall drop. Load was kept constant along the time while the breaking process moved from the mid span section towards the support edges. 

In the other, for inclined mesh (G-S-IM specimens), the grid geometry provided two concomitant resisting mechanisms: (i) shear resisting mechanism analogous to the case with straight mesh and (ii) axial resisting mechanism of the tows. Thus, fiber tows were subjected to shear and axial stresses at the same time. Due to the geometry, all tows were inclined and all of them contribute to the locking mechanism (there were no longitudinal and transversal fibers like in the straight case). As a result, all these tows of fiber were connected to a larger surface of concrete, in comparison with the straight orientation case, and more fiber quantity was involved in the failure mechanism. The observed failure was more progressive than the previous one, because of the concrete–mesh sliding process and the progressive tensile breaking of the mesh. These two mechanisms were allowed by the axial and shear response of the fiber. The corresponding global failure mode involved a larger FRP mesh–concrete connection area than the just the central section, resulting in higher load transfer capacity between these two phases and a higher load-bearing capacity. Complete interaction was only observed at the very beginning of the tests, whereas partial interaction mostly characterized these specimens’ performance (G-S-IM), even before the peak load.

Pictures of the different specimens after failure are included in Figure 5. In all cases, it was observed that failure was due to the debonding of the CFRP sheet from concrete. The debonding process started in the middle of the span and grew towards one of the edges. For the cases with no mesh connection (G), this debonding process was sudden and extended completely along one half of the slab. For the cases with mesh connection elements, the progression of the crack between the interfaces included a progressive formation of vertical bending cracks. These cracks were observed in the lateral part of the concrete block for the G-S-SM and G-SM cases. In contrast, G-S-IM specimens showed very thin microcracks that were smeared in a longer area. This result supports the idea of a larger and a more effective interlocking performance in the mesh–concrete connection, compared to the straight orientation. In this line, it can be noticed that the most loaded fabric tow in the case of a straight orientation covers a length in the longitudinal direction of the slab of 25 mm whereas this length is increased up to 400 mm for the 45⁰ inclined mesh case.

#### 4.2.2. Force vs. Vertical Displacement

Plots representing the force–vertical displacement response for all the tested cases are shown in Figure 6. Vertical displacement data were obtained by averaging both external potentiometers in the middle span. All the plots show a similar response characterized by a stiffer first linear part up to the first crack opening (approximately 2 kN of force), and a second part which is non-linear and captures the concrete cracking process and the progressive disconnection between the CFRP and concrete up to the maximum load. Subsequently, some cases showed a post-failure response with a clear load decrease. The lowest load-bearing capacities were related to the G-2 (4.87 kN) and G-S-SM-1 (8.0 kN) specimens, whereas the largest resistance was shown by the G-S-IM-1 (21.3 kN) and G-SM-1 (20.8 kN) specimens. Residual load-bearing capacity after failure is between 1 kN and 5 kN for all cases. This value is slightly higher for the cases including fabric connector and the greatest residual strength was registered for the cases with inclined mesh configuration (G-S-IM).

#### 4.2.3. CFRP–Concrete Sliding

All measurements of CFRP–concrete sliding show precisely the same behavior: the relative CFRP–concrete sliding at the extreme edges of the slabs started at the maximum load. Thus, there is no doubt, according to the experimental results, that the failure of all the specimens was due to the loss of connection between the CFRP thin sheet and the concrete block. It is also noticeable that in all cases, except the G-1 specimen, there is an asymmetric response, so the sliding took place in only one of the halves of the specimens. This is the expected response because random imperfections tend to cause the failure to start on one side or the other one, but not on both of them simultaneously. Anomalous symmetric response of G-1 case may be one of the reasons for its greater load-bearing capacity in comparison with G-2 case.

#### 4.2.4. CFRP Strain Distribution

The plots gathering the strain–time results of all tested specimens are displayed in Figure 7. Strain data were obtained directly via top and bottom strain gauges which were installed under the slab in the middle span (Figure 4b). The G-SM-1 top strain gauge measurements were not correct due to an incorrect adhesion of the strain gauges, so these were discarded. In all plots, the trend is linear-incremental for all strain gauges up to the maximum load. It can be observed that, initially, both the top and bottom strain gauges performed in tension until a certain point, corresponding to the maximum load and the specimen failure, from which the top ones started to show compressive strain. This fact indicates that the neutral axe may have moved from a position over the CFRP thin sheet to a position on the CFRP sheet height. Finally, at the end of the tests, the tensile bottom strain measurements were lower in accordance with the lower applied force. Maximum compressive strains were between 1000 με of G-S_SM cases and 2000 με of G-S-IM-2. It can be noticed that G-S-IM-1 case showed tensile strains at the top strain gage after maximum load. This fact proved the further contribution of the full CFRP section to resist the applied bending moment and justifies the second maximum peak of the load–displacement curve (Figure 6).

## 5. Calculations, Comparison and Discussion

### 5.1. Modal Analysis

Initially, it should be noted that the vibrational response of mechanical systems is influenced by the supports. As all the tested specimens had the same support, the results obtained from the experimental modal analysis (Section 4.1) can be used as comparative results. Thus, the results in Table 2 are not intended to produce an absolute vibration frequency value for the slabs, but can merely be used as a relative tool to find patterns that relate vibrational results with connection systems.

First, it is observed (see Table 2) that the G-S-SM-1 specimen showed an abnormally low vibration frequency (269 Hz) in comparison with the rest of the specimens, and its damping ratio was higher (4.93%) than the values for all other tests (between 1.2% and 2.5%). According to the results of the concrete compression tests (see Table 1), the strength of the concrete in this specimen was very low (16.4 MPa) in comparison with the other specimens (averaging 20 MPa). Therefore, it is surmised that there was an error in the concrete pouring process (some voids in the concrete were observed after a destructive test) that resulted in the concrete of this specimen being so different that it could not be used in a dynamic response comparison. From this point on, this specimen was rejected and was not considered for the remaining study of the modal analysis results.

Non-destructive experimental modal analysis tests showed that there was a certain correlation between the dynamical response and the connection system. The inclusion of a connection mesh led to an increase in the natural frequency (Table 2). The G samples had lower frequency values compared with G-SM (19.5%), G-S-SM (5.2%) and G-S-IM (8.1%). Results seem to indicate that adding sand to the gravel in the frictional mechanism was not beneficial (comparing G-S-SM with G-SM), but inclining the connection mesh had positive effects (comparing G-S-SM with G-S-IM). These non-destructive qualitative results are completely aligned with the destructive bending test results (see Section 5.2), proving that there is a certain relationship that requires further research to confirm the ability of the experimental modal analysis to relate the natural vibration frequency and particular connection systems in a quantitative way. In fact, general observations from the modal analysis indicate that this technique is highly dependent on the performance of individual materials. Thus, the connection system should not dramatically affect the results because the low excitation energy of this test results in a response in the elastic range, where the connection system is not very influential. It is not possible to see any pattern from the damping ratio.

In conclusion, including small-size particles for the roughness improvement of the CFRP–concrete contact surface (G-SM vs. G-S-SM) reduced the dynamic stiffness of the specimens and it was translated in a 12% lower frequency of vibration of the third bending mode. Thus, it is clear that small-size particles affect CFRP–concrete connection. In contrast, adding a straight mesh caused an increase of 20% of the vibration frequency (comparing G-SM and G cases), proving that the inclusion or not of this connection element may be comparatively assessed by experimental modal analysis. Finally, inclining the mesh has little effect (2.8% of the vibration frequency increase) supporting the idea that the orientation of the mesh has no effect in the lineal elastic response and only plays an important role at failure stage. 

### 5.2. Bending Tests

It should be observed that the obtained results show clear trends, despite the significant variability. The differences between sample results are related to the uncertainty of hand manufacture and the complexity of the interaction in the interfaces for shear transmission. This could present a problem for practitioners that pretend to use it at work, and it means that quality control must be mandatory to produce homogeneous specimens. Therefore, the analysis of the impact of the different connection techniques between the CFRP sheets and concrete is mostly qualitative.

Focusing on the overall comparison of the maximum load-bearing capacity, it can be observed that including the mesh connection (G-SM, G-S-SM and G-S-IM vs. G specimens) causes an average increase in the maximum load of approximately 60%. It can also be seen that placing sand, theoretically intended to increase the frictional response, provides the opposite result, reaching lower load-bearing capacities in comparison with the G-S-SM and G-SM specimens. This result is consistent with the observation from experimental modal analysis. 

Regarding the applied load, it is clear that incorporating a mesh (G-SM, G-S-SM and G-S-IM vs. G) allowed the slabs to withstand a residual strength after reaching the maximum peak and the corresponding load drop (see Figure 6). Hence, the novel connection proposal of including a flexible mesh had a positive effect, providing CFRP–concrete partial interlocking, even after a connection failure. This residual strength was greater for G-S-IM specimens (between 3 kN and 5 kN), supporting the idea that inclined mesh configuration brought more quantity of fibers to simultaneously collaborate even after the global failure of the structure. This fact translates to the practical implication of providing additional post-failure strength, so increased safety margin. Regarding the sand particles, it can be observed from the modal analysis (see Table 2) that incorporating them caused a decrease in the load-bearing capacity (G-S-SM vs. G-SM) that was also translated into a lower vibration frequency. However, it must be noted that, in the G-S-SM case, the slope of the force–displacement curve after the peak load was slightly positive, thus showing a weak hardening behavior and a partial recovering of the supported load. Moreover, the load drop in the G-S-SM case was less significant than in G-SM, which was approximately a 70% slighter drop. In summary, it seems that the inclusion of sand caused a premature partial failure of the connection, which remained with enough capacity to increase the withstand load again. In comparison, other connection types using mesh (G-SM and G-S-IM), in which the breaking damage was more extensive when occurring, showed a greater load drop and avoided the possibility of increasing the resisted load when increasing the displacement after the maximum load. 

Regarding the inclination of the mesh, it seems to have a positive effect on the load-bearing capacity of the proposed hybrid slabs (G-S-IM vs. G-S-SM). In addition, this case showed a better connection that translated to less bending cracks developing during tests and the failure mechanism that simultaneously mobilized almost all CFRP–concrete contact surfaces of one half of the slab. However, this case (G-S-IM) showed the second greatest load drop due to connection failure after no mesh (G) cases. Therefore, inclining the mesh helps to distribute the connection mechanism reaching higher loads, but also making the whole system more fragile, as the failure involves the full interface. 

Strain–time plots (see Figure 7) showed that the inclined mesh specimens (G-S-IM) achieved the greatest strain in CFRP, with a mean value of 5000 μm/m compared to 3500 μm/m for G-SM, and double that of the G specimens. The G-S-IM plots showed that the bonding of both materials performed as a single cross-section during the bending test for longer compared with the other specimens. Moreover, CFRP in G-S-IM performed for higher levels of load under tensile stresses than the rest of specimens, which means that the loss of adherence with concrete was more difficult. For all cases, when the connection was lost (maximum load), the top part of the CFRP suddenly started to work under compression. From this point on, the bending movement was no longer distributed according to Navier’s hypothesis. Overall, it can be ascertained that inclined mesh helps to uniformly distribute the connection between concrete and CFRP over a wider contact area than the other solutions. Thus, it is the preferred connection system from among the newly presented ones, including flexible mesh. 

Summarizing the effect of the key parameters on the direct results of the bending tests, it can be stated that including a flexible fabric connector (G-SM vs. G) increased the load-bearing capacity and brought residual strength. In contrast, including small-size particles in the CFRP–concrete contact (G-SM vs. G-S-SM) reduced the load-bearing capacity but allowed some post-peak strength because of higher frictional effect after contact failure. Finally, inclining the connection mesh (G-S_MS vs. G-S-IM) increased the load-bearing capacity and increased the residual strength but made the failure process more fragile. All these effects are related with the largest quantity of fibers simultaneously contributing (and simultaneously failing) to the connection strength of G-S-IM cases. 

#### 5.2.1. Evolution of the Position of the Neutral Axis

Figure 8 includes the neutral axis vs. vertical displacement at mid-span plots. As per the previous section, the vertical displacement was calculated with the average data from both external potentiometers. The position of the neutral axis was calculated from the measurements of the two strain gauges and by imposing the hypothesis of strain compatibility to the section. As previously indicated, the top strain gauge of the G-SM-1 specimen had no useful values, and it was not possible to calculate the position of the neutral axis for this case.

The neutral axis is referenced from the bottom part of the cross-section. When the neutral axis position was over 35 mm, the CFRP sheet was subjected to tensile stresses, and the concrete below the axis was supposed to be cracked. Conversely, when the axis was below 35 mm, the CFRP sheet was partially compressed. This is shown in Figure 7 which looks at the change in the signs of top strain measurements. 

The corresponding positions of the neutral axis during the linear-elastic branch (under 2 kN in Figure 6) are in a span of values between 40 mm and 70 mm. Thus, all the CFRP material and the mesh connection were subjected to normal tensile stresses at this moment. This position moved upwards at the first loading steps, but with the beginning of the non-linear load, the decrease was associated with the progressive cracking of the concrete (see Figure 6). The neutral axis position started to move downwards until reaching the maximum load, when the neutral axis position dropped down from a value over 35 mm to a value below this threshold (see Figure 8). Thus, the connection failure correlates in time with the sudden movement of the neutral axis that left the top part of the CFRP and the connection mesh subjected to normal compression stresses. Therefore, the CFRP–concrete connection was assured while the neutral axis was over 35 mm and it was subjected to tensile stresses. When the CFRP sheet became compressed, local buckling failure was possible due to the narrow thickness of the sheet. This phenomenon was clearly observed in the G specimens due to the total disconnection between the CFRP and concrete (Figure 5).

The post-maximum load response, after the neutral axis drop, was characterized by an almost constant position value of the neutral axis (between 20 mm and 30 mm). This means that after the failure of the interface, a residual friction between the CFRP, the cracked concrete and the mesh still contributed a little, as was shown in Figure 6.

Analyzing the effect of the studied parameters, it was observed that including the mesh (G vs. G-SM) allowed to stabilize the position of the neutral axis between 40 mm and 50 mm for a long period of the test up to the maximum load, whereas the cases without mesh showed a clear downward movement of the position of the neutral axis from the very beginning. Among the cases with meshes, no clear tendency is observed, so the response of the position of the neutral axis seems not to depend on the inclination of the mesh.

#### 5.2.2. CFRP–Concrete Equivalent Shear Stress 

Figure 9 shows the CFRP–concrete connection shear stress vs. vertical displacement plots. As per the previous sections, vertical displacement was calculated by averaging the data from the external potentiometers. The CFRP–concrete shear stress assumed an equivalent uniform distribution of the compressive/tensile force to be transmitted between the bottom and top parts of the section. This was calculated by dividing the normal total compressive or tensile force in the mid-span section by the CFRP–concrete idealized contact surface in half of the length of the slab (length × width = 1 m × 0.4 m = 0.4 m^2^). This total compressive/tensile force was calculated from the strain distribution and the corresponding force equilibrium.

The overall shape of the plots in Figure 9 was similar to those in Figure 6. Nevertheless, they show more clearly the interface performance of both materials. For example, it can be seen that after reaching the maximum shear stress, the diagram goes down almost to a value of 0 for the G cases, indicating that no connection mechanism remained after reaching the maximum load-bearing capacity. In contrast, for the G-S-IM and G-SM cases, low frictional stress (between 0.05 and 0.10 MPa) from the rough surfaces of the materials kept an almost constant small value. Finally, the G-S-SM case showed that the level of shear stress was very low from the beginning of the test (maximum value of 0.16 MPa for G-S-SM-2 specimen), showing a weak bond between the concrete and the CFRP with the addition of sand particles. 

The particular analysis of the study variables showed that including a flexible fabric connector (G vs. G-SM) increased the residual shear strength (from 30 kPa to 50 kPa). The same effect is observed when inclining the connection mesh (G-S-SM vs. G-S-IM) that doubled the residual shear strength. Finally, adding small-size particles to the CFRP–concrete interface (G-SM vs. G-S-SM) clearly reduced the shear strength (from 0.37 MPa to 0.11 MPa in average) although slightly increased the residual shear strength (from 40 kPa to 65 kPa as average value).

#### 5.2.3. External Energy 

Table 3 presents the accumulated external energy at peak load, that maximum accumulated external energy and the increment of the later respect to the former one for every specimen. Last column shows the increment of the total accumulated energy respect to the G cases taken as reference. External energy was calculated from the force and vertical displacement at the loaded section (mid-span). Results showed that a significant amount of energy can still be dissipated after reaching the peak load for the cases including flexible fabric connector (G-SM, G-S-SM and G-S-IM) due to the progressive breaking of the fibers. In addition, considering that the connection area of the cases with inclined mesh is greater than for the straight mesh ones (see Section 4.2.1), it is confirmed that more relative energy dissipation was possible for the cases with straight mesh due to the progressive breaking of transversal fabric tows. In the case of inclined mesh, a greater amount of fabric is simultaneously contributing, so at failure, there was little amount of unloaded fabric to dissipate more energy after reaching the maximum load. Nevertheless, this simultaneous contribution of G-S-IM cases justifies this is the connection system that more external energy dissipates (350% more than G cases and double of the comparable cases with straight mesh). Dissipated energy is an indirect measure of the damage capacity and resilience of the structure. In this regard, the inclusion of the flexible mesh connector enables more energy to dissipate (approximately 3-fold more), and the performance is better still if the mesh is inclined with respect to the direction of the nerves.

## 6. Conclusions 

After performing an experimental modal analysis and eight bending tests on CFRP sheet–concrete hybrid slabs with four different connection alternatives, including the novel approach of embedding a flexible fiber mesh, the following conclusions were reached:

The structural bending response of this type of slab showed three stages. For small loads (approximately 2 kN in these tests), the specimens performed according Navier’s hypothesis with a linear elastic response of the cross-section. For higher loads, the concrete started to crack and the load–displacement response tended to be non-linear. The existence of interlocking elements (gravel, sand and mesh) helped to keep the complex shear transmission mechanisms between materials. Finally, a peak load was reached, and a sudden failure occurred due to the loss of the connection between the concrete and CFRP sheet. The load dropped down and both materials were separated leaving a residual shear mechanism from the friction of the rough surfaces. Thus, the structural response of this novel connection system between thin FRP sheets and concrete is qualitatively described. 

Gravel particles bonded to the CFRP sheet surface provided a superficial connection with concrete. The performance of this connection was fragile and presented the lowest values of load-bearing capacity. This connection system as a stand-alone is not recommended for practical applications.

The novel inclusion of a flexible glass-fiber mesh directly bonded onto the CFRP, as a connector between the composite sheet and the concrete, allowed an increase in the load-bearing capacity of the slabs, produced residual strength after connection failure and increased the stiffness of the vibrational response of the slab specimens. 

A qualitative relationship between the vibrational response obtained from the experimental modal analysis and the load-bearing capacity was observed, although further research is required to confirm this fact and to progress its quantification.

The inclusion of sand particles bonded to the inner bottom surface of the CFRP had the unexpected effect of reducing the load-bearing capacity and the cohesion between the concrete and the CFRP sheet, although the post-peak residual strength was slightly increased. 

The inclination of the mesh clearly contributed to uniformly distributing the shear forces in the interface of the materials. This is likely to be because more fibers were oriented in a direction closer to the shear effort to be resisted, compared to the case of a straight position. Therefore, the inclined mesh seems to be the best choice of the two orientations of this novel connection strategy.

All CFRP sheet–concrete hybrid slabs failed when the neutral axis moved into the CFRP area (under 35 mm) causing the top part of the CFRP and the connection of the glass fiber mesh, when existing, to be compressed and to definitively separate from the concrete. Even thin CFRP can suffer buckling, as was observed.

The amount of cumulative energy corresponds with a qualitative estimation of the capacity of the slabs to dissipate energy and resist damage. The inclined mesh solution was confirmed to be the most competitive one. 

In conclusion, the main design lines of thin hybrid FRP–concrete slabs based on flexible fiber fabric connectors are provided as results of this study. Two of them have to be highlighted: using aggregates larger than 5 mm to provide roughness to the inner FRP surface and positioning the fiber fabric in an inclined orientation to distribute the load-transfer mechanism after reaching maximum load along a greater area. Finally, this study opens the door to a wide research field on the characterization of the structural response of these type of structures (or other ones based on the novel proposed flexible distributed connectors), including the study of different materials, geometries or adhesives. These results also put the basis for the development of analytical and numerical models of these systems. 

## Figures and Tables

**Figure 1 polymers-13-02862-f001:**
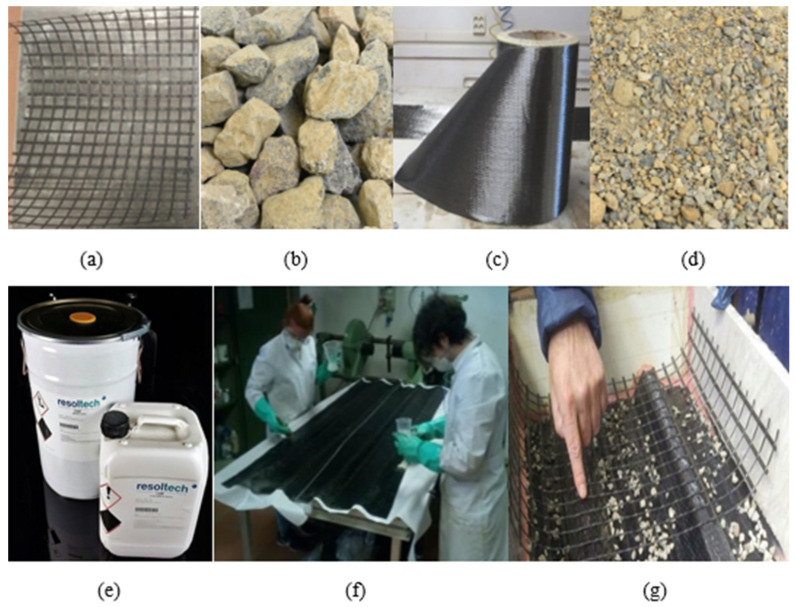
Materials and manufacturing; (**a**) Mapegrid G220 mesh, (**b**) FIP 300/500 carbon fiber, (**c**) 5–12 mm gravel, (**d**) 0–4 mm sand, (**e**) Resoltech 1200–1204 resin, (**f**) Hand layout CFRP sheet, (**g**) Mesh connection before casting.

**Figure 2 polymers-13-02862-f002:**
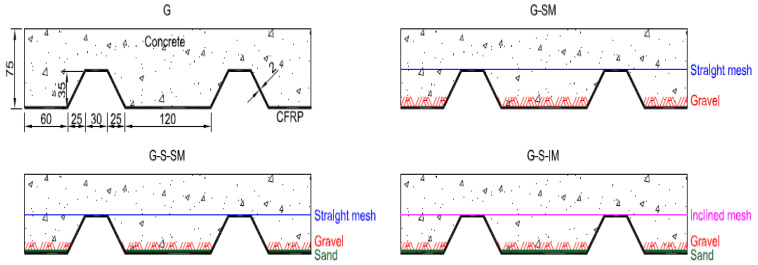
Connection system definitions and specimen cross-section dimensions (in mm).

**Figure 3 polymers-13-02862-f003:**
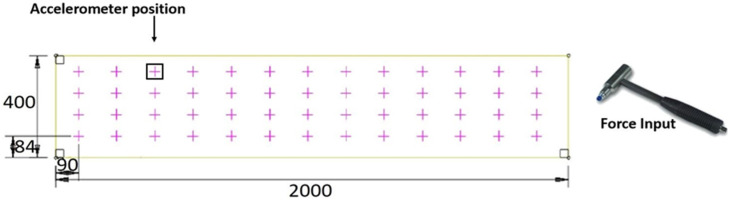
Modal analysis setup (dimensions in mm).

**Figure 4 polymers-13-02862-f004:**
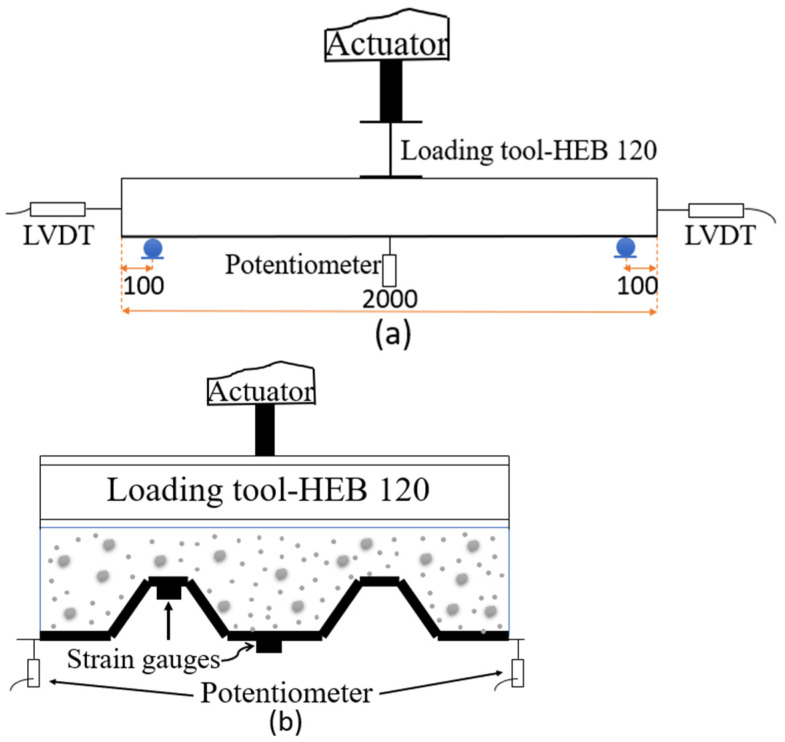
Test setup from longitudinal and transverse view (dimensions in mm).

**Figure 5 polymers-13-02862-f005:**
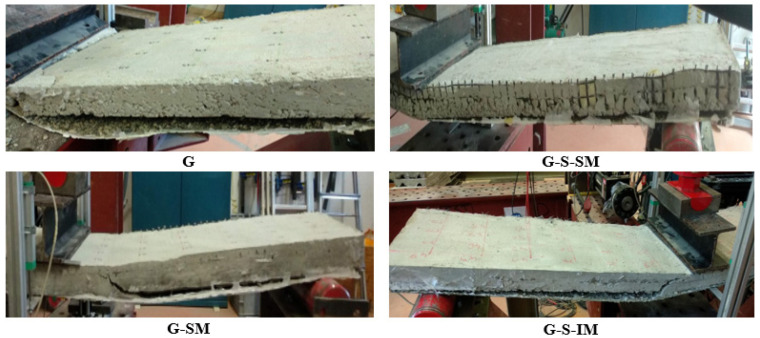
The failure mode of debonding for all specimen types.

**Figure 6 polymers-13-02862-f006:**
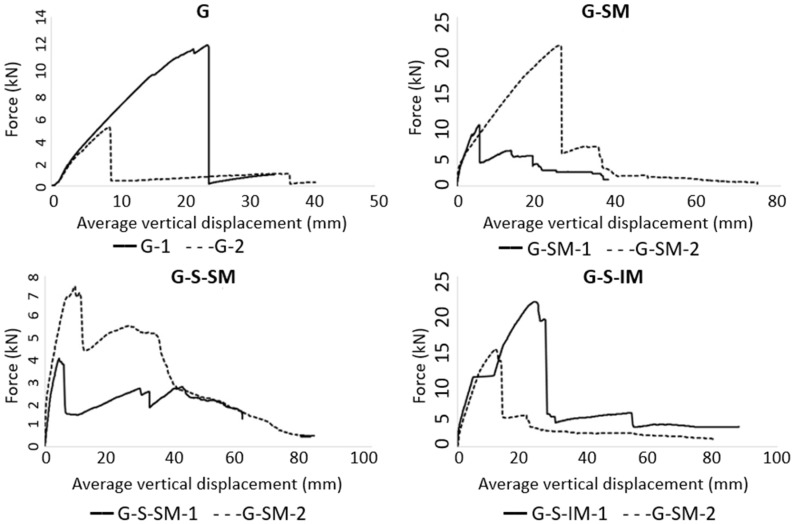
Force–average vertical displacement plots.

**Figure 7 polymers-13-02862-f007:**
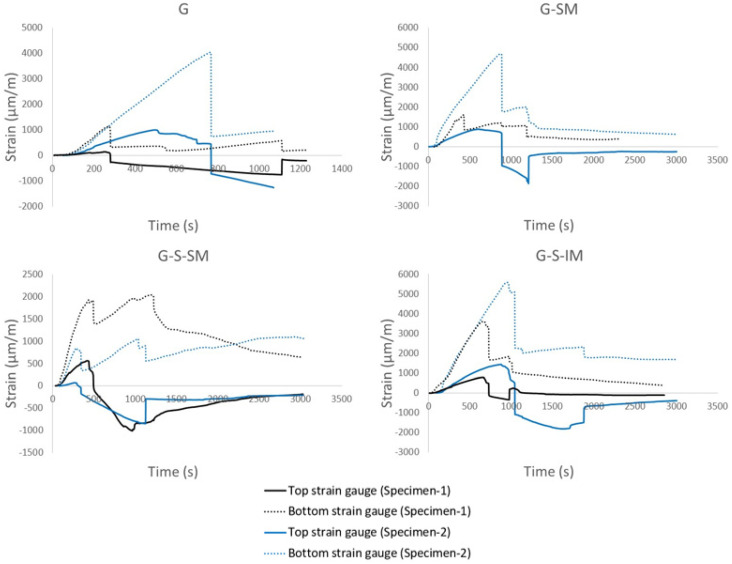
Strain_time plots.

**Figure 8 polymers-13-02862-f008:**
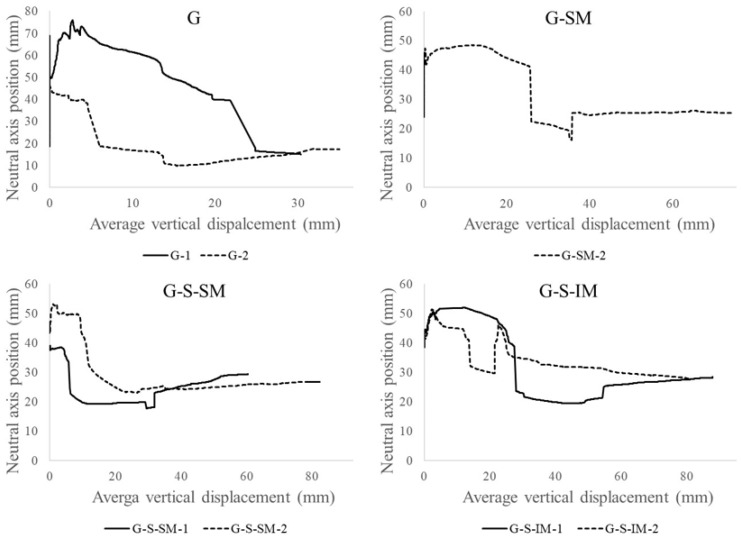
Neutral axis-average vertical displacement plots.

**Figure 9 polymers-13-02862-f009:**
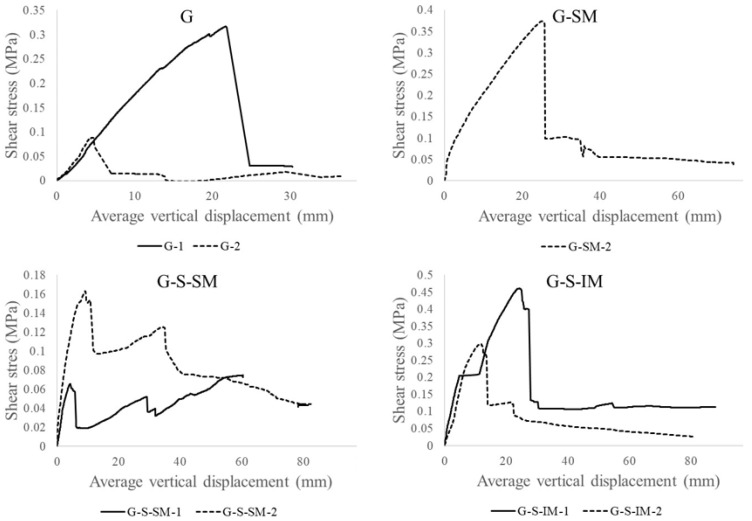
Shear stress-average vertical displacement plots.

**Table 1 polymers-13-02862-t001:** Details of the specimens.

Specimen	Gravel	Sand	Straight Mesh (0°)	Inclined Mesh (45°)	Concrete Average Compressive Strength (MPa)
G-1	Y	-	-	-	19.0
G-2	Y	-	-	-	20.8
G-SM-1	Y	-	Y	-	20.0
G-SM-2	Y	-	Y	-	22.6
G-S-SM-1	Y	Y	Y	-	16.4
G-S-SM-2	Y	Y	Y	-	20.4
G-S-IM-1	Y	Y	-	Y	20.0
G-S-IM-2	Y	Y	-	Y	20.9

**Table 2 polymers-13-02862-t002:** Vibration modes, frequencies and damping ratios.

	Third Bending Mode		
Specimen	ω (Hz)	$\bar{\omega }$ (Hz)	ζ (%)
G-1	312	307.0	1.22
G-2	302		1.69
G-SM-1	358	367.5	1.71
G-SM-2	377		1.18
G-S-SM-1	269	323.0	4.93
G-S-SM-2	323		1.35
G-S-IM-1	349	332.0	1.28
G-S-IM-2	315		2.52

**Table 3 polymers-13-02862-t003:** Cumulative external energy before and after reaching the maximum load. Relative increment respect to the energy at peak load. Average energy relative increase respect to G cases.

Specimen	E_peak_ (J)	E_max_ (J)	ΔE_max-peak_ (%)	ΔE_max-G_ (%)
G-1	168.0	175.4	3.8	-
G-2	24.6	46.0	87.0	
G-SM-1	62.3	255.6	310.3	225.6
G-SM-2	345.6	465.2	34.6	
G-S-SM-1	22.4	213.0	850.9	128.0
G-S-SM-2	71.9	291.8	305.8	
G-S-IM-1	408.0	644.6	58.0	350.0
G-S-IM-2	195.4	351.7	80.0	

## Data Availability

MDPI Research Data Policies.

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
