# Peer review of "Flexible Fiber Fabric for FRP–Concrete Connection of Thin Hybrid Slabs"

_polymers, 2021, doi:10.3390/polym13172862_

Round 1

Reviewer 1 Report

Comments

This paper studied FRP-concrete connection of thin hybrid slabs. The outcome of the paper is interesting however, there are several aspects that need to be improved. The reviewer can only recommend for publication if the author satisfactorily address the following major comments in the revised version.

  1. Please check carefully the references. Some of them shows “Error! Reference source not found.”
  2. The properties of the FRP material should be provided.
  3. The load transfer mechanism and failure mode of the specimens should be described clearly.
  4. The research questions and justification of selected parameters should be highlighted.
  5. The results and discussion was written in general. It is expected that the effect of key parameters should be discussed.
  6. The novelty of the study should be highlighted more clearly at the end of introduction section. How this study is different from the published study in literature?
  7. How the outcome of this study will benefit researchers and end users? This need to be highlighted in introduction or end of conclusion.
  8. The recent investigation on FRP-concrete composites should be discussed in introduction section to improve the background study. Recently, the FRP-concrete concept was used in manufacturing hybrid slab [Ref: Bending behaviour of precast concrete slab with externally flanged hollow FRP tubes], column repairing [Ref: State-of-the-art of prefabricated FRP composite jackets for structural repair] and wall system [Ref: Axial compression behaviour of all-composite modular wall system]. Suggest to include them in introduction section with proper citations to improve the background study.

I would be happy to see the revised version to understand how these comments are being addressed.

Reviewer 2 Report

The work entitled “Flexible fiber fabric for FRP-concrete connection of thin hybrid slabs” by Mahboob et al. analyzed different connection systems and proposed a novel approach of embedding a flexible fiber fabric as a superficially distributed connector between concrete and the fiber-reinforced polymer. The work is of quality and is very detailed with many experiments being conducted about different connection systems.

The authors did a good job describing the methodology and the materials that would be employed in the work but it is still required some improvement in the results and discussion section. Data must be more detailed and the discussion must also be criticized/supported by literature. There is a need for improving these sections prior to publication.

In detail:

  • The introduction must be reduced, and the novelty must be highlighted along the information.
  • The English writing must be improved.
  • There are several formatting errors that should be fixed.
  • There are too many figures.
  • In many sections there is barely discussion of the acquired data.

Round 2

Reviewer 1 Report

Thanks for addressing the comments. One last comment, the author names are incorrect for reference 1.

[1] A. Sharda et al., “Bending behaviour of precast concrete slab with externally flanged hollow FRP tubes,” Eng. Struct., vol. 23, 642
no. April, pp. 1244–1258, 2021, doi: 10.1016/j.jestch.2020.02.006.

Please correct the author's name for this reference.

Reviewer 2 Report

The authors have implemented the necessary alterations in the manuscript and it is now ready for publication. They have made significant changes in the discussion and in the introduction, which were the main sections requiring the attention of the authors.